# Enhanced Viral Activity in the Surface Microlayer of the Arctic and Antarctic Oceans

**DOI:** 10.3390/microorganisms9020317

**Published:** 2021-02-04

**Authors:** Dolors Vaqué, Julia A. Boras, Jesús Maria Arrieta, Susana Agustí, Carlos M. Duarte, Maria Montserrat Sala

**Affiliations:** 1Institut de Ciències del Mar-Consejo Superior de Investigaciones Científicas (ICM-CSIC), Passeig Marítim de la Barceloneta 37–49, 08003 Barcelona, Spain; juliaboras@gmail.com (J.A.B.); msala@icm.csic.es (M.M.S.); 2Centro Oceanográfico de Canarias (Instituto Español de Oceanografía, IEO), Farola del Mar 22, Dársena Pesquera, 38180 Tenerife, Spain; jesus.arrieta@ieo.es; 3Red Sea Research Center (RSRC), King Abdullah University of Science and Technology (KAUST), Thuwal 23955, Saudi Arabia; susana.agusti@kaust.edu.sa (S.A.); carlos.duarte@kaust.edu.sa (C.M.D.)

**Keywords:** prokaryotes, viruses, virus-mediated mortality, surface microlayer, subsurface water, Arctic and Antarctic Oceans

## Abstract

The ocean surface microlayer (SML), with physicochemical characteristics different from those of subsurface waters (SSW), results in dense and active viral and microbial communities that may favor virus–host interactions. Conversely, wind speed and/or UV radiation could adversely affect virus infection. Furthermore, in polar regions, organic and inorganic nutrient inputs from melting ice may increase microbial activity in the SML. Since the role of viruses in the microbial food web of the SML is poorly understood in polar oceans, we aimed to study the impact of viruses on prokaryotic communities in the SML and in the SSW in Arctic and Antarctic waters. We hypothesized that a higher viral activity in the SML than in the SSW in both polar systems would be observed. We measured viral and prokaryote abundances, virus-mediated mortality on prokaryotes, heterotrophic and phototrophic nanoflagellate abundance, and environmental factors. In both polar zones, we found small differences in environmental factors between the SML and the SSW. In contrast, despite the adverse effect of wind, viral and prokaryote abundances and virus-mediated mortality on prokaryotes were higher in the SML than in the SSW. As a consequence, the higher carbon flux released by lysed cells in the SML than in the SSW would increase the pool of dissolved organic carbon (DOC) and be rapidly used by other prokaryotes to grow (the viral shunt). Thus, our results suggest that viral activity greatly contributes to the functioning of the microbial food web in the SML, which could influence the biogeochemical cycles of the water column.

## 1. Introduction

The sea surface microlayer (SML), with a thickness of <1000 µm [1], is a vast habitat covering 70% of the earth’s surface and is regarded as a fundamental constituent in the air–sea exchange processes and in biogeochemical cycling [2]. This neustonic realm is called the skin of the ocean and is considered an extreme environment with specific chemical and biological properties as compared to subsurface waters [1]. Due to environmental forces, nutrients and dissolved organic matter (DOM) are accumulated in the SML, together with viruses, bacteria and archaea (prokaryotes), microalgae, and protists [1].

The SML habitat offers both advantages and disadvantages for viruses and microorganisms. On the one hand, it is exposed to more intense solar radiation, higher variations of temperature, salinity gradients, toxic organic substances, and heavy metals than in the subsurface waters (SSW) [3,4,5]. On the other hand, the SML could be enriched with natural and anthropogenic organic material, which favor the development of SML-bound microbial communities [3,6,7]. Marine viruses, prokaryotes, and other microorganisms (i.e., phytoplankton, small picoeukaryotes) present in the SML originate in underlying surface waters and can achieve very high abundances in the SML ([8] and references therein). All of them are adsorbed into air bubbles rising to the surface [9] and/or stick to organic particles that are transported to the SML via bubble scavenging [10].

The probability of viral infection and lysis of prokaryotes has been shown to increase within the SML due to the higher density of both viruses and hosts [8]. This would promote the leaching of dissolved organic matter from lysed cells, favoring the growth of other heterotrophic microorganisms [11,12]. However, it has been reported that extreme environmental conditions may affect the viral life strategy [13] and the physiological state of hosts [14]. Thus, high solar irradiance in the SML could lead to lysogeny as a viral infection strategy [15].

The distribution patterns of viruses and microorganisms in the SML and the relationship between their abundances and activities with the environment are poorly understood and often show divergent results, exhibiting either higher or lower abundances in different systems when comparing the SML relative to the SSW [8]. In addition, most studies investigating the role of viruses and their activity in the SML have been conducted in temperate systems, such as the Mediterranean Sea [14], and subtropical systems, such as Halong Bay, Vietnam [13]. In these systems, the SML tends to be enriched in organic molecules, mineral nutrients, and metals relative to the SSW, while viral and prokaryotic abundances and prokaryotic heterotrophic production oscillate between comparable and twice as high in the SML relative to the SSW [13,14].

The paucity of knowledge of the viral dynamics, their activity, interaction with environmental factors, and functioning in the microbial loop in the SML is even more significant in polar aquatic systems [8]. So far, almost no study assessing viral dynamics and mediated mortality on prokaryotes and the consequent release of organic carbon from the lysed cells was registered for polar system SML. Only one report is available for the Central Arctic [16], displaying the presence of smaller particles, between 20 and 60 nm, in the virioneuston compared to the usual size range of marine viruses, ranging between 20 and 200 nm [17]. During summer, in the SML of Arctic and Antarctic systems, the inflow of the melted sea ice, as well as the strong stratification and limited mixing caused by the overlying fresher layer resulting from ice melting, may favor the enrichment of viruses, microorganisms, and organic compounds. In addition, polar SML viral and microbial communities are subjected to stronger gradients of salinity and ambient temperature, as well as to episodes of strong wind and higher UV radiation along with extended photoperiods than those in lower latitudes. Furthermore, the ranges of temperature (higher in the Arctic) and other environmental variables such as salinity, organic nutrients, and inorganic nutrients may differ between Arctic and Antarctic waters [18,19], possibly resulting in different dynamics of the viral and microbial communities in the SML of these two polar areas.

We aimed to study the impact of viruses on prokaryotic communities in the SML and subsurface waters (SSW) of Arctic and Antarctic systems during summer. We hypothesized that in both polar areas, the input of organic and inorganic nutrients from ice melting in the SML could enhance the growth and abundance of prokaryotes and virus-mediated mortality, despite being highly influenced by high UV irradiance and wind. To achieve our goal, we sampled viral and microbial communities in the SML and SSW during two polar cruises, ATOS1 (European Sector of the Arctic Ocean) and ATOS2 (Antarctic Peninsula Sector of the Southern Ocean). We assessed viral abundance and virus-mediated mortality in prokaryotes, abundance and production of heterotrophic prokaryotes, and abundances of heterotrophic and phototrophic nanoflagellates. In addition, we measured several physicochemical factors (temperature, salinity, dissolved organic carbon (DOC), inorganic nutrients as well as atmospheric UV radiation and wind speed, that could influence these virus–prokaryote interactions.

## 2. Material and Methods

### 2.1. Sampling Sites and Strategy

The ATOS1 cruise took place from 27 June to 28 July 2007. Nineteen stations were sampled in the west and northern Greenland Sea and Arctic Ocean (68.5°–80.9° N, 2.6° W–19.5° E; Figure 1A) close to Svalbard Island. In the ATOS2 cruise, (28 January to 25 February 2009), we visited 17 stations around the Antarctic Peninsula located in the Bransfield Strait and the Weddell and Bellingshausen Seas (61.0°–69.5° N, 51.5°−76.1° W; Figure 1B). Both cruises were conducted on board the R/V BIO-Hespérides (for more details see [19,20,21]).

Environmental parameters, microbiological abundances, and prokaryotic production were measured at all visited stations during both cruises in the SML and SSW layers, except temperature, salinity, and DOC, which were only measured in the SSW for Antarctic waters (ATOS2) (Figure 1). Viral production, virus-mediated prokaryotic mortality, and lysogeny were determined at 9 stations in the Arctic Ocean (6, 9, 12, 15, 23, 33, 39, and 43; Figure 1A) and at 6 stations in Antarctic seawaters (2, 5, 7, 10, 13, and 17; Figure 1B) in both layers.

The surface microlayer (SML) water was sampled under calm sea conditions from a rubber boat deployed 2 km away from the research vessel in order to avoid contamination of the samples from the vessel’s influence. The SML samples were collected using a glass plate sampler [22], which had been previously cleaned with acid overnight and rinsed thoroughly with ultrapure water (MQ-water). To quantify any procedural contamination, we collected field SML blanks by rinsing and collecting 0.5 L of ultrapure water. We used a glass plate of a 975 cm^2^ surface area, and about 100 dips were required to collect 500 mL of the SML water. SSW samples were collected by hand at 0.1 m depth in an acid-cleaned plastic carboy from the same site as the SML samples. For chemical and microbiological parameters, we collected respectively 1 L and 600 mL from the SML and 2 L from the SSW layers.

The enrichment factor (EF) for chemical and biological variables was assessed as the ratio of the concentration or rate of the respective parameter in the SML to that in the SSW. An EF of >1.0 indicated an enrichment in the SML relative to the SSW; an EF of <1.0 indicated a depletion in the SML relative to the SSW.

### 2.2. Physicochemical and Atmospheric Parameters

Temperature in the SML and SSW was measured immediately after sampling on the rubber boat using a calibrated thermometer. Thus, this temperature corresponds to that of the collected sample, similar to that of the surrounding air in direct contact with the SML, but may slightly differ from the actual temperature in the SML. Samples for salinity were taken and stored refrigerated in a cool box to be later assessed in the laboratory of the R/V BIO-Hespérides with a salinometer (Portasal Guildline 8410-A). Duplicate 10 mL samples for determining dissolved inorganic nutrient (PO_4_ and NO_3_+NO_2_) concentrations were kept frozen until analysis in a Bran-Luebbe AA3 autoanalyzer (3 months after sampling, back to the laboratory of the Institut Mediterrani d’Estudis Avançats (IMEDEA-CSIC), following standard spectrophotometric methods [23]. NH_4_ concentration was determined spectrofluorometrically onboard within 1 h of collection using a Shimadzu spectrofluorometer [24]. Samples for DOC analyses were filtered through a GF/F filter and 10 mL aliquots transferred to duplicate glass ampoules, pre-combusted at 450 °C for 5 h, sealed under flame, and stored until analysis in the laboratory. DOC analyses were performed on a Shimadzu total organic carbon (TOC)-5000 or TOC-Vcsh following high-temperature catalytic oxidation techniques [25]. Standards provided by D.A. Hansell and W. Chen (University of Miami) of 2 mmol L^−1^ and 44 mmol L^−1^ TOC were used to assess the accuracy of the estimates. UV radiation and wind speed were automatically measured by a Weatherlink Vantage Pro. Davis Co. meteorological station located on board the R/V BIO-Hespérides (located 18.5 m above sea level). Wind speed was transformed to a standard height of 10 m by means of a power law with a 0.11 exponent [26]. UV (290–390 nm) values were obtained every 10 min with a UV Davis 6490 sensor. Its spectral response matches very closely the erythema action spectrum. UV data were displayed as an UV index.

### 2.3. Viral and Microbial Abundance

Subsamples (2 mL) for virus and prokaryote abundances were fixed with glutaraldehyde (0.5% final concentration) at 4 °C for 15–30 min and then quick-frozen in liquid nitrogen and stored at −80 °C, as described previously [27,28]. Counts were made on a FACSCalibur (Becton & Dickinson, Franklin Lakes, NJ, USA) flow cytometer in the ICM-CSIC laboratory (up to 6 months after sampling). Virus samples were diluted with TE-buffer (10:1 mM Tris:EDTA), stained with SYBR Green I, and run at a medium flow speed [27], with a flow rate of 58–64 µL min^−1^. Different groups of viruses were determined in bivariate scatter plots of green fluorescence of stained nucleic acids versus side scatter [29]. Depending on their fluorescent signal, viruses were classified as showing low (V1), medium (V2), or high (V3) fluorescence, which corresponds to their content in DNA. Presumably, V1 and V2 fractions are mainly attributed to bacteriophages, and V3 to viruses of phytoplankton [29]. Prokaryotic samples were stained with dimethyl sulphate (DMSO)-diluted SYTO13 and run at low speed using 0.92 µm yellow-green latex beads as an internal standard [30].

In situ nanoflagellate abundances were determined by epifluorescence microscopy (Olympus BX40−102/E at 1000×), with a blue wavelength excitation filter (BP 460–490 nm) and barrier filter (BA 510–550 nm), and with an ultraviolet excitation filter (BP 360–370 nm) and barrier filter (BA 420–460 nm). Subsamples (30 mL) were taken at the SML and SSW, fixed with glutaraldehyde (1% final concentration), filtered through 0.6 µm black polycarbonate filters, and stained with 4,6-diamidino-2-phenylindole (DAPI) at a final concentration of 5 µg mL^−1^ [31]. Phototrophic nanoflagellates (PNF) could be distinguished from heterotrophic nanoflagellates (HNF) under blue light, and the presence of plastidic structures in PNF could be observed with red fluorescence. At least 20–100 HNF and 20–100 PNF were counted per sample and separated by size classes of ≤5 µm and >5 µm.

### 2.4. Prokaryotic Heterotrophic Production

In situ prokaryotic heterotrophic production (PHP) was estimated from radioactive ^3^H-leucine incorporation [32], with the modifications established for the use of microcentrifuge vials [33]. Samples of 1.2 mL were dispensed into four 2 mL vials plus two TCA-killed control vials. Next, 48 µL of a 1 µM solution of ^3^H-leucine was added to the vials, providing a final concentration of 40 nM. Incubations were run for 3–4 h and stopped with TCA (50% final concentration). Next, the tubes were centrifuged for 10 min at 16,000 × *g*. Liquid was carefully aspirated with a Pasteur pipette connected to a vacuum pump. Pellets were rinsed with 1.5 mL of 5% TCA, vortexed, and centrifuged again. The supernatant was removed again, and 0.5 mL of scintillation cocktail was added. The vials were counted in a Beckman scintillation counter. The PHP was calculated according to the equation:PHP = Leu × CF (µg C L^−1^ day^−1^)
where Leu is the ^3^H-leucine incorporation (pmol L^−1^ h^−1^) and CF is the conversion factor (1.5 kg C mol Leu^−1^ [34]).

### 2.5. Viral Production and Rate of Lysed Prokaryotes

The virus reduction approach was used to determine viral production (VP) and prokaryote losses due to phages [35]. This approach aims to measure the VP in incubations, in which the initial concentration of viruses has been reduced in relation to the in situ viral abundance, yet maintaining the prokaryotic in situ concentration. Thus, the probability of the virus–bacteria encounter and new infections is reduced. It is assumed that the VP observed during the incubation time is a result of infections prior to incubation, that no new infections occur, and that both filtration and incubation do not induce lysogenic prokaryotes. This method also distinguishes between the production of lytic phages (viral lytic production (VPL)) and temperate phages (viral lysogenic production (VPLyso)) [36]. Lysogeny was detected with lysis induction by mitomycin C. Although this agent does not induce the lytic cycle in all prophages under certain conditions [37], this method is widely used, and therefore the results obtained can be compared to those of other studies. To perform the VP measurements, 0.5 L of the seawater from the SML and 1 L from the SSW layers were pre-filtered through a 0.8-µm-pore-size cellulose filter (Whatman) to remove grazers (e.g., nanoflagellates and ciliates) and then concentrated by a tangential flow cartridge (0.22 µm pore size, VIVAFlow 200) to obtain 40 mL of prokaryote concentrate. Water collected from the 0.22 µm filtrate was processed using a cartridge with a 100 kDa molecular mass cutoff (VIVAFlow 200) obtaining virus-free water. A mixture of virus-free water and prokaryote concentrate (in 4:1 *v*/*v* proportion) was prepared and distributed into four sterile 50 mL falcon plastic tubes. Two of the tubes were kept as controls (giving a viral lytic production), while mitomycin C (Sigma-Aldrich, St. Louis, MO, USA) was added (1 µg mL^−1^ final concentration) to the other two tubes as the inducing agent of the lytic cycle in prophages, giving the total viral production (lytic and lysogenic). All falcon tubes were incubated in a thermostatic chamber simulating in situ temperature, in the dark, for 12 h. Viruses and prokaryotes from viral production incubations were subsampled every 4 h and counted by flow cytometry, as described above, back to the lab in the ICM-CSIC. Calculations of viral lytic and lysogenic production (VPLyso) were made quantifying the difference between averaged viral production and viral lytic production duplicates according to [38] (VP = VPL + VPLyso). Since part of the prokaryotic abundance is lost during the concentration process, the VPL and VPLyso were multiplied by the corresponding prokaryote correction factor [36], which is obtained by dividing the prokaryote abundance in situ by the prokaryote abundance at time 0. The correction factor ranged from 0.2 to 10 in our study, enabling the comparison of the VP from different incubations. The number of viruses released by a prokaryote cell (burst size (BS)) was estimated from VP incubations as in [20,39]. Thus, the increase in viral abundance during short time intervals (4 h) was divided by the decrease in prokaryotic abundance over the same time period. We assumed that the PHP and viral decay during this time interval were negligible. The obtained BS ranged from 12 to 59 viruses per prokaryote in Arctic and from 50 to 126 viruses per prokaryote in Antarctic SML and SSW samples. These values are similar or higher to the average BS obtained before for Arctic (BS = 45 [40]) and Antarctic (BS = 50 [41]) waters.

The rate of lysed prokaryote cells (RLC) was calculated by dividing the VPL by the BS, as described by Guixa-Boixereu et al. [42]: RLC (cells lysed mL^−1^d^−1^) = VPL/BS. We also estimated the rate of released carbon (µg C L^−1^ d^−1^) from the lysed cells using the carbon-to-volume relationship [43] derived from the data of Simon and Azam [44]: pg C cell^–1^ = 0.12 × *V*^0.7^, where *V* is the volume of prokaryote cells in μm^3^. Here, we used a cell average volume of 0.066 µm^3^ prokaryote^−1^ reported for Antarctic waters [45]. Finally, from the RLC, we calculated the percentage of lysed prokaryote standing stock cells (PSS) expressed as percentage virus-mediated mortality (%VMM):%VMM (d^−1^) = RLC × 100/PSS.

The percentage of lysogenic cells (%Lysogeny) was calculated similarly to %VMM using the lysogenic viral production (VPLyso) that resulted from the induced prophages by mitomycin C to lytic bacteriophages:%Lysogeny (d^−1^) = VPLyso/BS × 100/BSS

### 2.6. Data Analyses

The Shapiro–Wilk *W*-test was used to check the normal distribution of data, and data were logarithmically transformed prior to analyses, if necessary. Pearson correlation and regression analyses were applied to assess relations among different biotic and environmental parameters. Differences in physicochemical and biological variables between the SML and the SSW in each system were tested by the two-tailed Student’s *t*-test paired data. Finally, to estimate differences of physicochemical and biological parameters between the Antarctic and Arctic SML and SSW, we carried out one-way ANOVA analyses. All statistical analyses were performed with Kaleidagraph 4.1.3 (Reading, PA, USA) and the JMP 8.0 (© SAS Institute Inc., Cary, NC, USA) programs.

## 3. Results

Values of environmental, viral, and microbial parameters for the SML and SSW for both polar systems are shown in Table 1 and Table 2.

### 3.1. Environmental Parameters

In the Arctic, water temperature and salinity in the SML and SSW varied from −0.9 to 5.5 °C and from 31.2 to 34.5 (Table 1), respectively. However, only temperature showed significantly higher values in the SML (Table 1). Both temperature and salinity in the SML were strongly correlated with the corresponding SSW values (*p* < 0.00001, Table 3). In Antarctic waters, temperature and salinity were only measured in the SSW. Temperature varied from −1.1 to 3.2 °C and achieved significantly lower values than in the Arctic (Table 1 and Table 4), whereas salinity showed significantly higher values in Antarctic (33.9 ± 0.1) than in Arctic (32.7 ± 0.2) waters (Table 1 and Table 4).

The dissolved organic carbon (DOC) concentration in the Arctic did not show significant differences between the SML and the SSW, achieving the highest value in the SML (808 µmol C L^−1^) around Svalbard and the lowest in the SSW (71.4 µmol C L^−1^) in the Greenland Sea transect (Figure 1A, Table 1). In Antarctic waters, the DOC concentration, only measured in the SSW, ranged from 46.9 µmol C L^−1^ in the Weddell Sea to 249.1 µmol C L^−1^ in the Bellingshausen Sea (Table 1) and was much lower than in the Arctic SSW (Table 4). Only the Arctic SML presented a higher NH_4_ concentration compared to the SSW (EF = 1.7 ± 0.5; Table 1, Figure 2A). In Antarctic waters, significantly higher values of the NH_4_ concentration were observed in both layers compared to the Arctic (Table 1 and Table 4). Finally, DOC and all inorganic nutrient concentrations were correlated between the SML and SSW layers (Table 3).

UV index radiation varied significantly from 5.4 ± 0.06 (range of 1.6–8.9) around Svalbard (Arctic) to 2.5 ± 0.4 in the Antarctic waters (from 0.7 in the Bellingshausen Sea to 5.2 in the Weddell Sea; Table 1 and Table 4).

Wind speed (WS) achieved similar average values in the Arctic and Antarctic (5.0 ± 0.4 m s^−1^ and 5.5 ± 0.5 m s^−1^, respectively; Table 1 and Table 4), although the range was slightly wider in the Antarctic (2.6–9.9 m s^−1^) than in the Arctic (1.2–7.9 m s^−1^) during the sampling days. Wind speed was negatively correlated with PO_4_ and the DOC only in the SML in the Arctic (Appendix A). Consequently, the enrichment factor (EF) of these variables decreased with increasing wind speed (PO_4_ at WS > 6 m s^−1^: 0.9 ± 0.3 µmol P L^−1^; WS = 5–6 m s^−1^: 1.3 ± 0.1 µmol P L^−1^; WS <5 m s^−1^: 1.4 ± 0.6 µmol P L^−1^; and DOC at WS > 6 m s^−1^: 0.9 ± 0.1; WS = 5–6 ms^−1^: 1.3 ± 0.3; WS < 5 m s^−1^: 1.6 ± 0.5) (Table 5).

### 3.2. Viral and Microbial Parameters

Almost all viral and microbial variables tended to be higher in the SML as compared to the SSW, except for phototrophic nanoflagellate abundance (Table 2, Figure 2C,D) in the Arctic. The number of parameters presenting significant differences between the SML and the SSW were higher in Arctic (viral abundance, virus prokaryote ratio, prokaryotic heterotrophic production, rate of lysed cells, percentage of virus-mediated mortality, and phototrophic nanoflagellate abundance; Figure 2C) than in Antarctic (viral abundance, prokaryote abundance, viral lytic production, and phototrophic nanoflagellate abundance; Figure 2D) waters.

### 3.3. Viral and Microbial Abundances and Prokaryotic Heterotrophic Production

In both cruises, viral abundance (VA) was much higher (*p* < 0.05) and more variable in the SML than in the SSW (Table 2, Figure 2C,D). The average EF ranged from 1.7 ± 0.6 to 2.4 ± 1.1 in the Greenland Sea transect (T) and around Svalbard (S), respectively, and from 1.8 ± 0.7 to 4.1 ± 1.7 in the Bellingshausen Sea and Weddell Sea, respectively (Table 2). When comparing both polar systems, there were not significant differences between the VA (Table 4). However, higher values were observed in Antarctic waters (Table 2), ranging from 0.04 to 32.1 × 10^7^ virus mL^−1^, compared to a range from 0.1 to 10.8 × 10^7^ virus mL^−1^ in the Arctic (Table 2). The majority of viruses corresponded to the fraction of bacteriophages (V1 and V2), reaching values, on average, between 70% and 90% of total viruses in the SML of Arctic and Antarctic waters, respectively (data not shown).

Prokaryote abundance (PA), although slightly higher in the SML than in the SSW, was only different between these two layers (*p* < 0.05) in Antarctic waters (Table 2, Figure 2C,D). The average EF for PA ranged from 1.0 ± 0.4 to 1.3 ± 0.9 in the transect and Svalbard, respectively, and from 1.0 ± 0.05 to 1.4 ± 0.7 in the Bransfield Strait and Bellingshausen Sea, respectively (Figure 2C,D). Conversely to viral abundances, PA was higher (*p* < 0.01) in Arctic than in Antarctic waters (Table 4). In both cruises, VA and PA were correlated between layers (*p* < 0.00001; Table 3). Furthermore, VA and PA showed a similar distribution in the SML, resulting in a significant correlation (Arctic: *r* = 0.621, *p* < 0.001; Antarctic: *r* = 0.517, *p* < 0.05; Appendix A) between those two variables, suggesting that these prokaryote communities were the dominant hosts for viruses.

The average virus-to-prokaryote ratio (VPR) tended to be higher in the SML than in the SSW (Figure 3A,B), although differences between both layers were only significant for the Arctic Ocean (Table 2, Figure 2C,D). In Antarctic waters, the average VPR was very variable, ranging from 18.1 ± 5.4 to 43.9 ± 32.2 in the SML in the Bransfield Strait and Weddell Sea, respectively, and from 11.1 ± 2.2 to 46.0 ± 27.6 in the SSW in the Bransfield Strait and Bellingshausen Sea, respectively (Table 2, Figure 3A,B). In both polar systems, VPR was correlated between layers (*p* < 0.0001, Table 3), and VPR value both for the SML and for the SSW were much higher in Antarctic with respect to Arctic waters (*p* < 0.001; Table 4). Furthermore, in the Arctic, the VA was positively correlated with temperature, DOC, PO_4_, and SiO_4_ (Appendix A) and the PA with the same variables except for temperature. In Antarctic waters, both variables were negatively correlated with PO_4_ (*r* = −0.593, *p* < 0.01) and SiO_4_ (*r* = −0.595, *p* < 0.01) (Appendix A). For the SSW, the PA was correlated with NO_3_+NO_2_ and SiO_4_ in Arctic waters (Appendix A) and the VA was negatively related with salinity and positively with DOC in Antarctic waters (Appendix A).

Heterotrophic and phototrophic nanoflagellates (HNF and PNF) achieved higher abundances in Arctic than in Antarctic waters both in the SML and in the SSW (Table 2), although HNF did not show statistically significant differences between layers and between polar systems (Figure 2C,D, Table 4). In the Arctic, HNF of ≤ 5 µm, considered the main prokaryote grazers, achieved similar average percentage values (~80%, Table 2) as the total HNF in the SML and SSW. In contrast, in Antarctic waters, we obtained lower values at HNF of ≤ 5 µm in the SSW (29–83%) than in the SML (71–92%). It is noticeable that PNF abundance in the SML in the Arctic registered significantly lower values (9.2 ± 3.3 × 10^3^ cells mL^−1^) than in the SSW (10.1 ± 1.4 × 10^3^ cells mL^−1^). The EF for PNF in Antarctic waters was significantly higher (3.1 ± 1.7) than in Arctic waters (0.8 ± 0.2) (Table 4). Finally, we did not find any correlation between viral abundance and HNF and PNF (Appendix A).

Prokaryotic heterotrophic production (PHP) in the SML showed similar values for Arctic (3.8 ± 0.9 µg C L^−1^ d^−1^) and Antarctic (3.9 ± 1.0 µg C L^−1^ d^−1^) waters. Conversely, significantly higher PHP estimates were registered in the SSW in Antarctic (3.9 ± 1.1 µg C L^−1^ d^−1^) than in Arctic (1.3 ± 0.3 µg C L^−1^ d^−1^) waters (Table 2 and Table 4). PHP in both cruises was correlated between the SML and the SSW (*p* < 0.001; Table 3), and the EF, although similar for both polar systems (1.5 ± 1.0 in Arctic and 1.2 ± 1.0 in Antarctic waters; Table 2), was significantly greater than 1 in the Arctic Ocean (Figure 2C).

Furthermore, in the SML of the Arctic, PHP was negatively correlated with the VPR and HNF, while in the SSW, it was positively related with PNF (Appendix A). Likewise, for Antarctic waters, PHP was strongly positively correlated with the VA and VPR in the SML and negatively with PO_4_ and NO_3_+NO_2_ and with the PA and HNF in the SSW (Appendix A).

Wind speed negatively affected the VA (*r* = −0.512, *p* < 0.01; Appendix A) in the SML of the Arctic Ocean, resulting in increasing EF with decreasing wind speed, from 3.0 ± 1.5 at low to 0.9 ± 0.1 at high wind speed (Table 5). Similarly, in Antarctic waters, both VA and PA in the SML were negatively correlated with wind speed (*r* = −0.423, *p* < 0.01 and *r* = −0.492, *p* < 0.01, respectively; Appendix A), and at increasing wind speed, the EF decreased (Table 5). In addition, in the SML of Antarctic waters, UV radiation, likewise, negatively influenced the VA (−0.627, *p* < 0.01) and PHP (*r* = −0.607, *p* < 0.05; Appendix A). Moreover, only in the Antarctic underlying water (SSW), the VA and VPR were negatively correlated with both wind (*r* = −0.508, *p* < 0.01 and *r* = −0.406, *p* < 0.05, respectively) and UV radiation (*r* = −0.704, *p* < 0.01 and *r* = −0.715, *p* < 0.01, respectively) (Appendix A).

### 3.4. Viral Life Strategies (Lysis vs. Lysogeny)

Viral lytic production (VPL) was detected for almost all sampled stations, except at station 33 near Svalbard (Figure 1A and Figure 3C,D). Among the stations in both polar systems, the VPL varied by a factor of 10, both for the SML and the SSW, achieving the highest value in the SML of station 10 of the Bransfield Strait and station 7 in the Weddell Sea (Figure 1B and Figure 3D). Significantly higher values of the VPL in the SML compared to the SSW were registered only in Antarctic waters (Figure 2D and Figure 3D), but this was not always the case in Arctic waters (Figure 2C and Figure 3C).

Lytic infection, expressed as a percentage of virus-mediated mortality (%VMM), varied from undetected to 38.6% in the SML of Arctic waters (Figure 3E) and from 24.1% to 141.4% in the Bransfield Strait and Bellinghaussen Sea, respectively (Figure 3F). In the Arctic, the %VMM in the SML was significantly higher than in the SSW (Figure 2C, Table 2). Finally, significantly higher %VMM values in the SML and SSW were obtained in Antarctic with respect to Arctic waters (Table 4). The occurrence of lysogeny (% of lysogenic prokaryotes) was low in both polar systems. For the SML in the Arctic, lysogeny was observed at stations 6 and 20 and in Antarctic waters only at station 17 (Figure 1B and Figure 3E,F) and oscillated from 0.2% to 13.5% (Arctic) and 40.6% (Antarctic) (Figure 3C,D). In the underlying waters (SSW) of the Arctic, lysogeny was detected at different stations than for the SML (9, 12, and 33; Figure 1A and Figure 3E), fluctuating from 0.2% at station 12 to 2.2% at station 9 (Figure 3E). In the Antarctic SSW, lysogeny was detected in the Bransfield Strait at stations 10 and 13 and at station 7 in the Weddell Sea, ranging from 1.7% to 36.7% (Figure 3F). In both polar systems, lysogeny was lower than lytic infection (%VMM), except for station 33, where only lysogeny was detected (Figure 3E,F).

### 3.5. Mortality Rates

The rate of lysed prokaryote cells (RLC) varied in the SML from non-detectable to 1.4 × 10^6^ prokaryote mL^−1^ d^−1^ in Arctic waters and from 0.1 × 10^6^ to 1.5 × 10^6^ prokaryote mL^−1^ d^−1^ in Antarctic samples. For the SSW, it oscillated from non-detectable to 0.4 × 10^6^ prokaryote mL^−1^ d^−1^ and from 0.1 × 10^6^ to 0.5 ×10^6^ prokaryote mL^−1^ d^−1^ in Arctic and Antarctic waters, respectively (Table 2, Figure 3G,H), being significantly higher in Antarctic than in Arctic waters (Table 4). The corresponding carbon released varied in both layers around Svalbard from below the detection limit at station 33 to 24.9 µg C L^−1^ d^−1^ in the SML of station 23. In contrast, in Antarctic stations, minimal and maximal values were observed in the Bransfield Strait at station 10 in the SSW (0.9 µg C L^−1^ d^−1^) and in the Weddell Sea at station 7 in the SML (26.7 µg C L^−1^ d^−1^) (Table 2). The rate of lysed cells in the SML of both polar systems was positively related with viral and prokaryote abundances (Figure 4A–D).

In addition, the released organic carbon due to viral lysis in the SML was related to the DOC only in Arctic waters (no DOC data for the SML was available for Antarctica; Figure 4E,F). In contrast, we did not find any relationship between viral abundance and activity with prokaryote abundance or DOC concentration in the SSW (Figure 4A–F). The released carbon was, on average, 27 times higher in the SML than in the SSW in Arctic waters. Finally, wind speed (WS) negatively affected the RLC in the SML of Arctic (*r* = −0.798, *p* < 0.001; Appendix A) and Antarctic (*r* = −0.815, *p* < 0.01; Appendix A) waters. In this case, the EF for the RLC was greatly reduced in the Arctic under high wind speed conditions (from 47.9 ± 21.2 at low WS to 7.5 ± 3.4 at high WS; Table 5).

## 4. Discussion

In this study, we examined viral and microbial abundances and, for the first time, viral activity in the SML of Arctic and Antarctic seawaters. Our results indicated that there were no major differences between the SML and SSW physicochemical conditions for both polar regions. In contrast, higher values of viral and prokaryote abundances, as well as viral activity, were found in the SML as compared to the SSW. Our findings also suggested that in the SML, prokaryote communities were the dominant hosts for viruses. Therefore, the rate of lysed prokaryotes depended on viral and host abundances. Consequently, the cell carbon released in the SML contributed to the DOC pool concentration to a greater extent than that released by viral activity in the SSW.

The assessed SML temperature in the Arctic registered slightly but significantly higher values in the SML than in the SSW. Although we have to take these values with caution due to technical limitations and the evidence from other studies showing that temperature tends to be lower in the SML than in the SSW [46], the significant correlation between temperature and viral abundance only in the SML (*r* = 0.436, *p* < 0.05, Appendix A) suggests that increasing temperatures would contribute to the increase in viral abundance [47].

Almost all viral and microbiological variables tended to be higher in the SML than in the SSW (Table 2), and several of them were correlated between both layers (Table 3). Similar results were reported in different studies [8,13,14,48] despite the high variability of the SSW depth sampled, which oscillated from 0.1 m to >5 m. Viral abundance in the SML was, on average, ~2.0- to 3.0-fold higher than in the SSW, with the maximal EF up to 16-fold at station 23 in the west of Svalbard and 12 times at station 7 in the Weddell Sea. These values agree with earlier reports of high viral abundance enrichment (10–12-fold) in the SML in other systems [49], suggesting that it could be related to active transfer of viruses to the sea surface with bursting bubbles, or by atmospheric deposition [50]. Prokaryotes achieved on average lower enrichment factors (1.1–1.3) than viruses (1.7–2.4), according to values reported previously [14]. Furthermore, at some stations (6 of 16 in the Arctic and 4 of 19 in the Antarctic), the EF was lower than 1 for both VA and PA, as was also detected in some locations of the Mediterranean Sea [14] and in Halong Bay [13]. In addition, lower values in the SML than in the SSW (EF < 1) only for the PA were also found in the subtropical Atlantic Gyre, in the Western Mediterranean Sea [51], and in the Raunafjorden, Norway [52].

The type of devices used to sample the SML could affect the EF [53,54]. In our study, we used a glass plate, which, together with metal screens, seems adequate for sampling prokaryotes and viruses, as well as for investigating the structure of bacterial communities in the SML [12,53]. Indeed, Rahlff (2019, and references therein) [8] summarized results of EF for viral and prokaryote abundances obtained with different SML samplers for different marine ecosystems, and those were in the same order of magnitude.

Wind speed affects the level of the EF in polar SMLs, with strong wind speed disrupting the SML and mixing it with the SSW [55]. This is reflected in the reduced enrichment of the VA, PA, and the rate of lysed cells (RLC) under wind speeds of > 6 m s^−1^ (Table 5), which is consistent with reports for ocean waters elsewhere [56,57]. Our results also showed negative correlations between wind speed and viral and prokaryotic abundances, viral activity, and DOC concentration in the SML during both cruises (Appendix A).

UV radiation may also participate in decreasing the EF of viruses and microorganisms and their activities in the SML [14,58]. However, in our study, we did not detect a clear effect of UV on them in the SML in both systems, although UV radiation reached higher index values during the Arctic cruise than during Antarctic sampling (Table 1), where we generally found stormy weather. Nevertheless, significantly lower values of PNF in the SML in the Arctic suggested a negative effect of UV on them (Figure 2C), as is shown in [59], where photoinhibition occurred on the phytoneuston under natural summer light intensities. In contrast, in Antarctic waters, with a lower UV radiation index than in the Arctic, PNF showed, on average, higher values in the SML (Table 1 and Table 2, Figure 2D). UV radiation has been also reported to negatively affect viral abundance and infectivity, as well as damage viral and prokaryotic cells [60,61,62,63]. It could have contradictory effects, inducing the lytic cycle in temperate viruses or enhancing lysogeny [64], even though natural UV radiation was not always efficient to induce lysogeny in surface waters [65].

The lytic viral strategy dominated over lysogeny in both polar regions. Our findings agree with the results in Halong Bay [13], where almost always lytic production exceeded lysogeny and differed from results in the SML of Lake Superior [15], where lysogeny was the most common strategy. There are many contradictory assumptions dealing with the surface viral life strategy (lysis versus lysogeny), involving the trophic status of the system or the metabolic status and fitness of the host [66]. The SML of polar waters is subjected to larger variability (temperature, salinity, CO_2_, inputs, and deposition of compounds) as compared to that in lower latitudes and to that in deeper layers of the water column. However, the available data are insufficient to find why lysis would prevail over lysogeny.

The possible negative effect of UV on virus or prokaryote genetic material could be counteracted by the specific environmental conditions found in the neuston [2], which could favor the activity and composition of microbial communities. Thus, although we did not find significant differences for almost any physicochemical factor between the SML and the SSW, we did observe that EF > 1 for viral and microbial variables as the VA, PA, as well as RLC in both polar systems. Furthermore, in the SML, all these variables (mostly in the Arctic) correlated with nearly all inorganic nutrients and DOC concentrations (Appendix A), suggesting that these characteristics are favorable for maintaining viral and prokaryotic communities and for structuring the microbial food web. Indeed, only the SML supported higher mortality rates by viruses linked to the increase in the abundance of viruses and host cells (prokaryotes) or activity (Appendix A, Figure 4). These results agree with the findings shown for the SML in Halong Bay [13], where the frequency of infected cells (FIC) was significantly correlated with bacterial production and viral abundance. In contrast, these relationships were not observed in the underlying waters, suggesting that enrichment of viruses and hosts in the SML increases the encounter probability rate and the infection rates. As a consequence, a larger fraction of viral carbon lysates was delivered in the SML than in the SSW (6.1 ± 2.7 and 1.1 ± 0.8 µg C L^−1^ d^−1^, respectively, in the Arctic; 11.6 ± 4.3 and 3.6 ± 1.1 µg C L^−1^ d^−1^, respectively, in the Antarctic), which would be rapidly consumed by prokaryotes, enhancing the transfer of matter from prokaryotes via dissolved organic matter (DOM) and back to bacteria (the viral shunt) [67,68,69,70].

Finally, to complete the picture of the microbial food web in the SML, we expected that HNF abundances would show a relationship with prokaryote abundances or a clear enrichment in the SML (Table 2, Figure 2C,D), but this was not the case (Appendix A). Unfortunately, we did not either measure grazing rates in the SML nor, up to now, have they been ever measured. Nevertheless, in grazing estimates carried out in the same Arctic and Antarctic [19,20] areas, we obtained higher grazing than viral lysis rates in the SSW at stations around Svalbard (Arctic), while prokaryote mortality rates by viruses were more relevant than grazing in Antarctic waters. In the present study, viral activity in the SSW was higher in Antarctic than in Arctic waters (Table 4). At the same time, the %HNF of < 5 µm (i.e., the main prokaryote grazers [71]) was higher in the SSW in Arctic than in Antarctic waters (Table 2). Around Svalbard, we detected similar high average values of %HNF of < 5 µm in the SML and SSW (80.3% ± 4.5% and 83.4% ± 5.1%, respectively), suggesting a similar grazing activity in both layers. In Antarctic waters, we assessed lower average values of %HNF of < 5 µm in the SSW (67.1% ± 2.4%) than in the SML (82.4% ± 3.2%), suggesting higher grazing activity in the SML. This interpretation should be taken with caution and tested in future studies measuring simultaneously grazing and lytic prokaryotic rates to understand the role of HNF and viruses in shaping the structure and functioning of the microbial food webs in the SML.

## 5. Conclusions

In the SML of both polar systems, physicochemical factors (i.e., DOC and inorganic nutrients) mediated in sustaining viral and prokaryotic abundances and viral activity. In contrast, perturbations, such as strong wind, produced a decrease in viral and microbial abundances and viral activity, but still maintaining higher values in the SML than in the SSW. Furthermore, prokaryotes appeared as the main hosts for viruses only in the SML, being the released prokaryotic carbon due to viral lysis larger in the SML than in the SSW. This released organic matter was probably used by other prokaryotes to grow (the viral shunt). Finally, the detection in the SML of a high proportion of %HNF of < 5 µm in both polar systems suggested that grazers, together with viruses, might constitute a very active part of the microbial food web in this layer. Further studies are needed to confirm these results to provide a big picture of microbial food web functioning, its role in the biogeochemical cycles in the SML, and its influence on the water column.

## Figures and Tables

**Figure 1 microorganisms-09-00317-f001:**
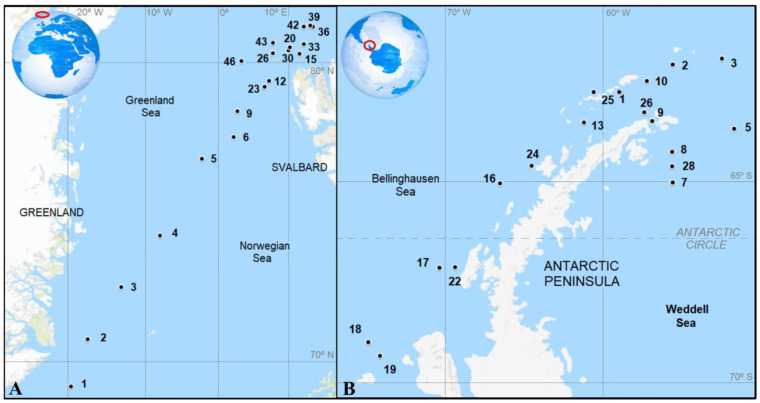
Map of sampled stations in the Arctic (Greenland Sea and around Svalbard) (**A**) and Antarctic waters (Bransfield Strait, Weddell Sea, and Bellingshausen Sea) (**B**) during ATOS1 and ATOS2 cruises, respectively.

**Figure 2 microorganisms-09-00317-f002:**
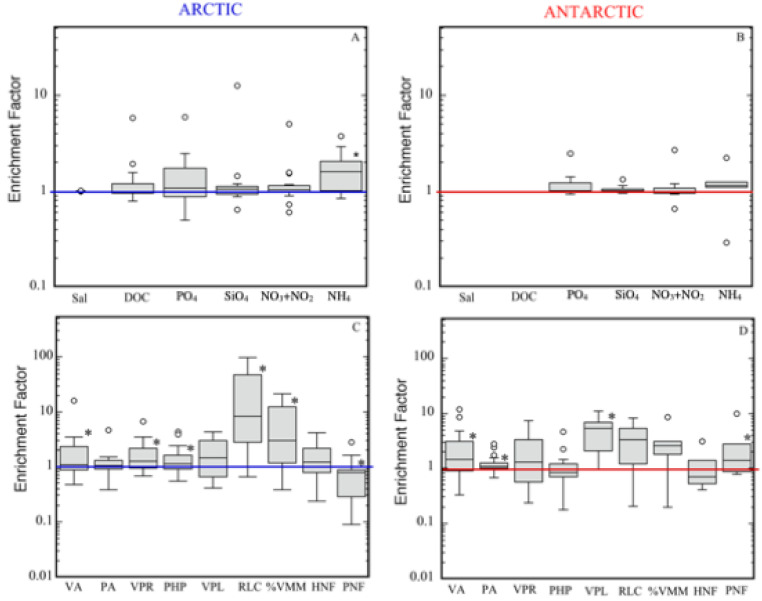
Box-whisker plots of enrichment factors in Arctic and Antarctic waters. Chemical variables (Sal: salinity; DOC: dissolved organic carbon; PO_4_: phosphate; SiO_4_: silicate; NO_3_+NO_2_: nitrite plus nitrate; and NH_4_: ammonia concentrations) (**A**,**B**). Microbial parameters (VA: viral abundance; PA: prokaryote abundance; VPR: virus prokaryote ratio; VPL: viral lytic production; RLC: rate of lysed cells; %VMM: virus-mediated mortality; HNF: heterotrophic nanoflagellate abundances; and PNF: phototrophic nanoflagellate abundances) (**C**,**D**). Horizontal lines within boxes indicate the median of the distribution, and the box limits are 25% quartiles of the data. The whiskers cover the entire data range, except for outliers (°), some of which are off-scale. * Significant EF values different than 1. The horizontal line in each figure corresponds to EF = 1.

**Figure 3 microorganisms-09-00317-f003:**
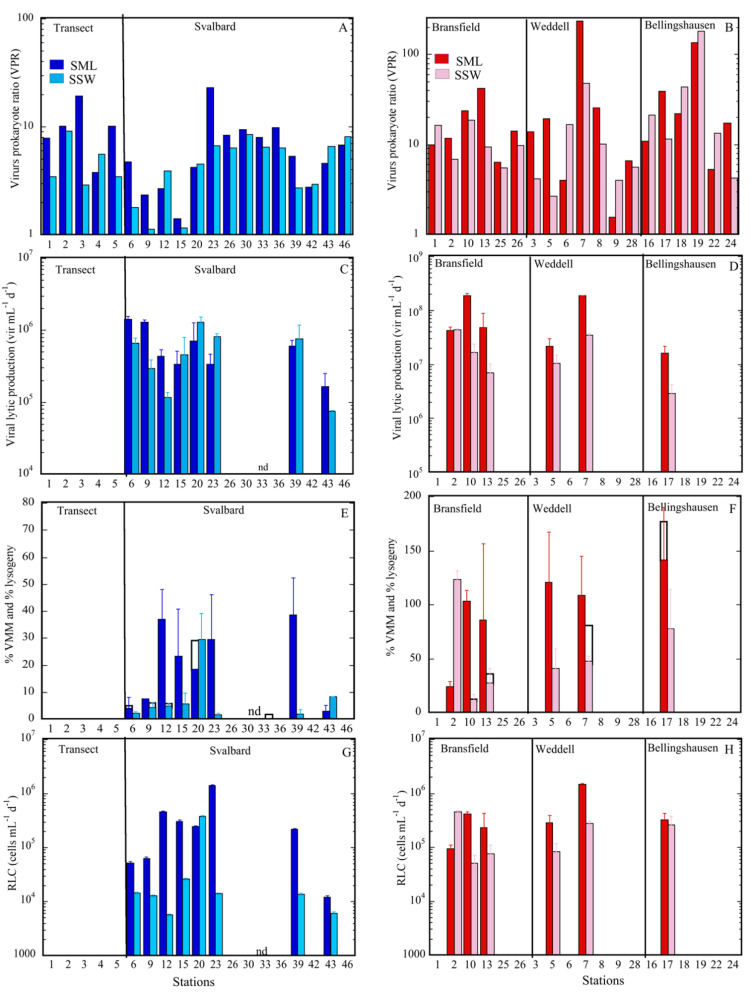
Viral parameters at the sampled stations of the Arctic and the Antarctic in the SML (dark-blue and red columns) and the SSW (light-blue and red columns) waters. VPR (**A**,**B**); VPL (**C**,**D**); lytic infection (%VMM, full columns), and %Lysogeny (empty columns) (**E**,**F**); and RLC (**G**,**H**). Error bars in each column indicate the maximum and minimum values of duplicates measured during the incubations, except for %Lysogeny (averages of total viral production—average VPL; see Material and Methods); nd: non-detectable. For acronyms, see Figure 2.

**Figure 4 microorganisms-09-00317-f004:**
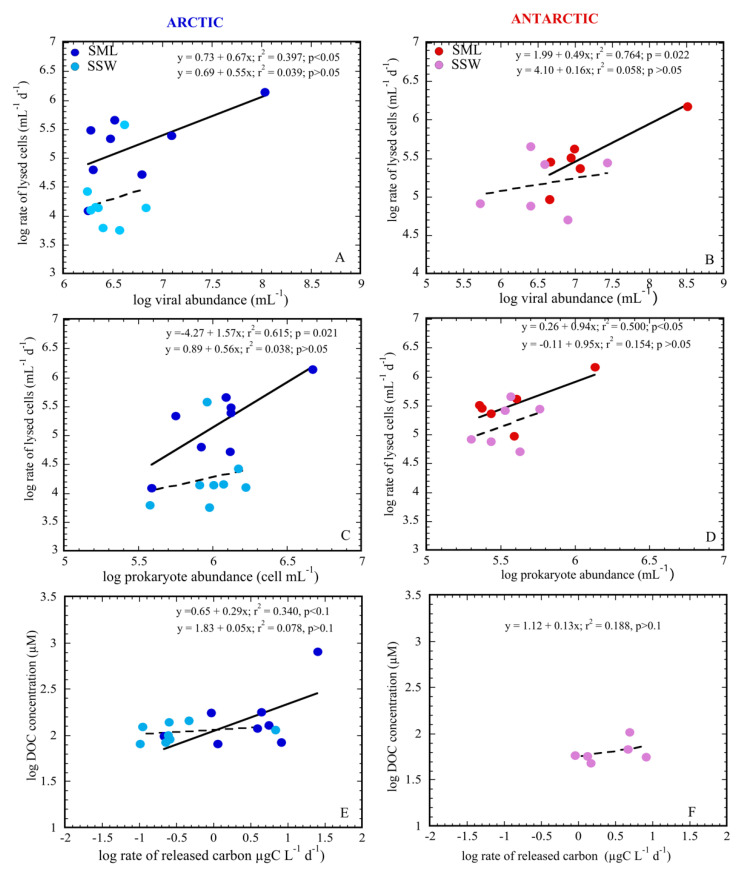
Regression analyses between the rate of lysed cells (RLC) with viral and host abundances, and the rate of released carbon with DOC in Arctic (**A**,**C**,**E**) and Antarctic (**B**,**D**,**F**) waters for the SML and SSW. Significant levels of *p* < 0.05 and of 0.05 < *p* < 0.1 were considered. Analyses and drawings were performed using the JMP 8.0 and Kaleidagraph 4.0 programs.

**Table 1 microorganisms-09-00317-t001:** Mean, minimal, and maximal values of physicochemical variables (VAR) of the surface microlayer (SML) and subsurface water (SSW) from Arctic (AR; T: transect; S: Svalbard; ATOS 1) and Antarctic (AN; Br: Bransfield; We: Weddell; Be: Bellingshausen; ATOS2) seawaters. EF: enrichment factor, in bold significant values (*p* < 0.05) different than 1. Temp: temperature; DOC: dissolved organic carbon; PO_4_, SiO_4_, NO_3_+NO_2_, and NH_4_: inorganic nutrients; *n*: number of values; nd: non-detected; -: no data. Wind: wind speed (m s^−1^) and UV index.

VAR	AR	SML	SSW	EF (SML/SSW)	*n*	AN	SML	SSW	EF (SML/SSW)	*n*
Temp	All	2.7 (−0.9–4.8)	2.1 (−0.9–5.5)	**1.2** (0.2–2.9)	19	All	-	1.1 (−0.4–3.2)	-	18
°C	T	3.7 (2.8–4.4)	2.6 (2.3–2.9)	1.4 (1.2–1.6)	5	Br	-	1.6 (−0.2–2.5)	-	6
	S	2.3 (−0.9–4.8)	2.0 (−0.9–5.5)	1.3 (0.2–2.9)	14	We	-	0.1 (−0.4–0.9)	-	6
						Be	-	1.7 (−1.1–3.2)	-	6
Salinity	All	32.6 (31.2–34.4)	32.7 (31.5–34.5)	1.0 (1.0–1.0)	19	All	-	33.9 (32.5–34.3)	-	18
	T	32.2 (31.3–33.4)	32.2 (31.5–33.4)	1.0 (1.0–1.1)	5	Br	-	34.2 (34.1–34.2)	-	6
	S	32.8 (31.2–34.4)	32.8 (31.5–34.5)	1.0 (1.0–1.0)	14	We	-	33.9 (33.4–34.3)	-	6
						Be	-	33.5(32.5–33.8)	-	6
DOC	All	144.2 (71.5–808.8)	98.3 (71.4–145.0)	1.4 (0.8–5.9)	18	All	-	72.3 (46.0–241.9)	-	18
µM	T	91.2 (71.5–111.1)	82.9 (71.4–92.5)	1.1 (0.9–1.4)	5	Br	-	58.6 (50.4–71.1)	-	6
	S	164.6 (79.2–808.8)	105.3 (80.4–145.0)	1.5 (0.8–5.9)	13	We	-	60.5 (46.0–103.5)	-	6
						Be	-	95.5 (47.9–241.9)	-	6
PO_4_	All	0.1 (0.0–0.9)	0.1 (0.0–0.3)	1.5 (0.5–6.0)	18	All	1.2 (0.5–1.6)	1.1 (0.3–1.6)	1.2 (0.9–2.5)	14
µM	T	0.1 (0.1–0.2)	0.1 (0.0–0.2)	1.3 (0.5–1.8)	5	Br	1.5 (1.3–1.6)	1.2 (0.6–1.4)	1.6 (1.0–2.5) 1.1(0.9–1.4) 1.1(1.0–1.2)	3
	S	0.1 (0.0–0.9)	0.1 (0.0–0.3)	1.6 (0.5–6.0)	13	We	0.9 (0.5–1,4)	1.0 (0.3–1.4)	1.1 (0.9–1.4)	5
						Be	1.3 (1.1–1.6)	1.2 (1.1–1.6)	1.1 (1.0–1.2)	6
SiO_4_	All	3.0 (0.5–27.3)	1.5 (0.6–3.6)	1.7 (0.6–12.7)	18	All	58.2 (40.2–68.5)	58.3 (39.6–71.1)	1.0 (1.0–1.3)	14
µM	T	1.1 (0.5–2.3)	1.1 (0.6–2.3)	1.0 (0.9–1.1)	5	Br	65.4 (63.5–68.5)	65.2 (51.9–71.1)	1.1 (1.0–1.3)	3
	S	3.7 (0.7–27.3)	1.7 (0.8–3.6)	1.9 (0.6–12.7)	13	We	49.0 (40.2–63.3)	50.5 (39.6–63.1)	1.0 (1.0–1.1)	5
						Be	62.4 (47.8–68.3)	60.3 (48.6–66.3)	1.0 (1.0–1.2)	6
NO_3+_NO_2_	All	0.7 (0.1–3.2)	0.9 (0.0–5.3)	1.3 (0.6–5.0)	18	All	15.0 (6.6–21.0)	15.6 (6.6–21.1)	1.1 (0.7–1.7)	13
µM	T	0.4 (0.2–0.5)	0.3 (0.0–0.6)	1.9 (0.9–5.0)	5	Br	18.3 (18.1–20.1)	17.0 (7.4–20.9)	1.6 (0.9–1.7)	3
	S	0.8 (0.1–3.16)	1.1 (0.1–5.3)	1.0 (0.6–1.2)	13	We	13.5 (6.6–21.0)	15.0 (6.6–21.0)	1.0 (0.9–1.1)	4
						Be	16.7 (10.0–20.7)	17.2 (15.3–21.1)	1.0 (0.7–1.2)	6
NH_4_	All	0.5 (0.2–1.2)	0.3 (0.1–0.7)	1.7 (0.8–3.8)	19	All	1.3 (0.1–4.6)	1.1 (0.1–3.1)	1.1 (0.3–2.2)	6
µM	T	0.5 (0.2–1.2)	0.4 (0.2–0.7)	1.2 (0.9–1.9)	5	Br	0.1	0.3 (0.2–0.3)	0.3	1
	S	0.4 (0.2–0.9)	0.3 (0.1–0.7)	1.8 (0.8–3.8)	14	We	0.3 (0.1–0.5)	1.2 (0.1–3.1)	1.1 (1.1–1.3)	2
						Be	2.8 (1.6–4.6)	1.8 (1.4–2.1)	1.1 (1.1–2.2)	3
**VAR**	**AR**	**Atmosphere**			***n***	**AN**	**Atmosphere**			***n***
Wind	All	5.0 (1.2–7.9)			19	All	5.5 (2.6–9.9)			19
m s^−1^	T	4.3 (1.2–6.8)			5	Br	5.7 (3.3–8.5)			6
	S	5.2 (1.8–7.9)			14	We	5.9 (3.5–9.9)			7
						Be	4.7 (2.6–7.9)			6
UV	All	5.4 (1.6–8.9)			13	All	2.5 (0.7–5.2)			11
index	T	-			-	Br	2.1 (1.4–3.9)			4
	S	5.4 (1.6–8.9)			13	We	3.5 (2.1–5.2)			4
						Be	1.6 (0.7–2.4)			3

**Table 2 microorganisms-09-00317-t002:** Mean, minimal, and maximal values of viral and microbial variables of the surface microlayer (SML) and subsurface water (SSW) of the Arctic (AR; T: transect of Greenland Sea; S: Svalbard; ATOS 1) and Antarctic (AN; Br: Bransfield strait; We: Weddell Sea; Be: Bellingshausen Sea; ATOS2) seawaters. EF: enrichment factor, in bold significant values (*p* < 0.05) different than 1. VA: viral abundance; PA; prokaryote abundance; VPR: virus prokaryote ratio; PHP: prokaryotic heterotrophic production; VPL: viral lytic production; %VMM: percentage of virus-mediated mortality (lytic infected cells); %Lysogeny: percentage of lysogenic infected cells; RLC: rate of lysed cells; RLC_C: rate of carbon released from lysed cells; HNF and PNF: heterotrophic and phototrophic nanoflagellates; %HNF < 5 µm: percentage of HNF smaller than 5 µm; *n*: number of values; nd: non-detected; -: no data.

VAR	AR	SML	SSW	EF	*n*	AN	SML	SSW	EF	*n*
VA	All	9.6 (1.3–108.3)	3.3 (1.2–6.8)	**2.2** (0.5–15.9)	19	**All**	25.7 (0.4–321.0)	8.1 (1.3–42.9)	**2.7** (0.3–11.7)	19
10^6^ mL^−1^	T	4.8 (1.5–11.7)	3.5 (1.2–6.7)	1.7 (0.5–3.4)	5	Br	6.4 (3.6–11.5)	4.3 (2.5–7.9)	1.8 (0.7–4.5)	6
	S	11.6 (1.3–108.3)	3.2 (1.6–6.8)	2.4 (0.7–15.9)	14	We	49.9 (0.4–321.0)	5.6 (1.5–27.4)	4.1 (0.3–11.7)	7
						Be	16.9 (3.0–22.5)	14.5 (1.3–42.9)	1.8 (0.4–4.8)	6
PA	All	1.0 (0.2–4.7)	0.8 (0.4–1.9)	1.2 (0.4–4.7)	19	All	0.5 (0.2–2.1)	0.4 (0.2–0.8)	**1.3** (0.7–2.8)	19
10^6^ mL^−1^	T	1.0 (0.2–1.5)	0.8 (0.4–1.9)	1.1 (0.4–1.5)	5	Br	0.4 (0.3–0.8)	0.4 (0.3–0.8)	1.0 (1.0–1.1)	6
	S	1.1 (0.4–4.7)	0.8 (0.4–1.7)	1.3 (0.5–4.7)	14	We	0.5 (0.2–1.4)	0.4 (0.2–0.6)	1.4 (0.8–2.4)	7
						Be	0.7 (0.2–2.1)	0.4 (0.2–0.7)	1.4 (0.7–2.8)	6
VPR	All	7.1 (1.2–23.0)	4.8 (1.1–9.1)	**1.7** (0.0–6.8)	19	All	34.1 (1.6–136.5)	22.8 (2.7–181.1)	2.1 (0.2–7.3)	19
	T	8.5 (1.2–19.5)	4.9 (2.9–9.1)	2.2 (0.0–6.8)	5	Br	18.1 (6.4–42.6)	11.1 (5.5–18.7)	1.8 (0.6–4.5)	6
	S	6.7 (1.4–23.0)	4.8 (1.1–8.5)	1.5 (0.7–3.4)	14	We	43.9 (1.6–236.2)	13.0 (2.7–47.7)	2.9 (0.2–7.3)	7
						Be	38.5 (5.3–136.5)	46.0 (4.2–181.1)	1.6 (0.4–4.1)	6
PHP	All	3.8 (0.4–18.6)	1.3 (0.3–5.1)	**1.5** (0.6–4.3)	19	All	3.9 (0.6–18.7)	3.9 (0.8–21.4)	1.2 (0.2–4.7)	19
µg C L^−1^ d^−1^	T	2.0 (0.5–3.9)	1.9 (0.3–5.1)	1.2 (0.8–1.6)	5	Br	2.1 (0.6–2.7)	5.1(0.8–21.4)	1.0 (0.2–2.2)	6
	S	3.4 (0.4–18.6)	2.1 (0.4–4.3)	1.6 (0.6–4.3)	14	We	4.2 (1.0–10.9)	3.1 (1.6–9.5)	1.1 (0.4–2.2)	7
						Be	5.3 (1.6–18.7)	3.1 (2.2–4.0)	1.5 (0.7–4.7)	6
VPL	All	0.07 (nd−0.1)	0.06 (nd−0.1)	1.9 (nd−4.4)	9	All	8.4 (2.2–18.9)	1.9 (0.3–4.4)	**5.4** (1.0–11.2)	6
10^7^ mL^−1^ d^−1^	T	-	-	-	-	Br	9.3 (4.2–18.9)	2.3 (0.7–4.4)	6.4 (1.0–11.2)	3
	S	0.07 (nd−0.1)	0.06 (nd−0.1)	1.9 (nd−4.4)	9	We	10.4 (2.2–18.7)	2.3 (1.0–3.5)	3.7 (2.1–5.4)	2
						Be	1.7	0.3	5.4	1
VMM	All	20.2 (nd−38.6)	7.3 (nd−29.6)	**6.9** (nd−17.4)	9	All	97.4 (24.1–141.4)	55.0 (11.9–123.5)	3.7 (0.2–8.7)	6
%	T	-		-	-	Br	71.2 (24.1–103.5)	54.4 (11.9–123.5)	4.0 (0.2–8.7)	3
	S	20.2 (nd−38.6)	7.3 (nd−29.6)	6.9 (nd−17.4)	9	We	114.7 (108.5–120.9)	44.5 (41.2–47.8)	4.2 (2.9–5.4)	2
						Be	141.4	78.2	1.8	1
Lysogeny	All	6.8 (nd−13.5)	0.9 (nd−2.2)	nd	9	All	40.3 (nd−40,3)	16.2 (nd−36.7)	nd	6
%	T	-	-	-	-	Br	nd	6.0 (nd−10.2)	nd	3
	S	6.8 (nd−13.5)	0.9 (nd−2.2)	nd	9	We	nd	36.7 (nd−36.7)	nd	2
						Be	40.3	nd	nd	1
RLC	All	0.3 (nd−1.4)	0.1(nd−0.4)	**27.2** (nd−99.4)	9	All	0.7 (0.1–1.5)	0.2 (0.1–0.5)	7.2 (0.2–8.3)	6
10^6^ mL^−1^d^−1^	T	-	-	-	-	Br	0.3 (0.1–0.4)	0.2 (0.1–0.5)	3.4 (0.2–8.3)	3
	S	0.3 (nd−1.4)	0.1 (nd−0.4)	27.2 (nd−99.4)	9	We	0.9 (0.3–1.5)	0.2 (0.1–0.3)	4.4 (3.5–5.4)	2
						Be	0.3	0.1	1.2	1
RLC_C	All	6.1 (nd−24.9)	1.1 (nd−6.8)	**27.2** (nd−99.4)	9	All	11.6 (1.7–26.7)	3.6 (0.9–8.1)	7.2 (0.2–8.3)	6
µg C L^−1^d^−1^	T	-	-	-	-	Br	10.8 (1.7–26.7)	3.5 (0.9–8.1)	3.4 (0.2–8.3)	3
	S	6.1 (nd−24.9)	1.1 (nd−6.8)	27.2 (nd−99.4)	9	We	15.8 (5.1–26.4)	3.2 (1.5–4.9)	4.4 (3.5–5.4)	2
						Be	5.7	4.7	1.2	1
HNF	All	2.6 (0.2–7.0)	2.5 (0.2–8.9)	1.6 (0.2–4.1)	19	All	1.4 (0.5–2.4)	1.5 (0.3–2.7)	1.3 (0.4–3.1)	6
10^3^ mL^−1^	T	4.3 (1.3–6.7)	3.9 (1.53–8.9)	1.4 (0.8–3.1)	5	Br	1.5 (1.0–2.4)	1.5 (0.3–2.5)	1.6 (0.4–3.1)	3
	S	1.8 (0.2–7.0)	1.7 (0.2–6.1)	1.6 (0.2–4.1)	14	We	1.9	1.9 (1.0–2.7)	0.7	2
						Be	0.5	1.0	0.5	1
%HNF < 5	All	80.5 (47.2–100)	83.5 (39.5–100)	1.0 (0.7–1.6)	19	All	82.4 (70.8–91.6)	67.1 (28.8–82.6)	1.5 (1.0–3.0)	6
	T	61.9 (42.4–88.3)	75.7 (39.5–98.9)	0.9 (0.7–1.6)	5	Br	83.2 (76.0–86.6)	59.3 (28.8–82.6)	1.7 (1.1–3.0)	3
	S	89.8 (47.2–100)	87.4 (63.7–100)	1.0 (0.7–1.3)	14	We	70.8	72.2 (70.7–73.7)	1.0	2
						Be	91.6	80.1	1.1	1
PNF	All	9.2 (0.2–44.3)	10.1 (1.7–19.0)	**0.8** (0.1–2.8)	19	All	3.5 (0.9–5.1)	2.0 (0.5–3.1)	**3.1** (0.8–9.8)	6
10^3^ mL^−1^	T	2.8 (0.2–7.5)	4.8 (1.7–7.9)	0.6 (0.1–1.0)	5	Br	3.8 (2.4–5.0)	2.1 (0.5–3.1)	4.0 (0.8–9.8)	3
	S	12.4 (0.5–44.3)	12.2 (5.7–19.0)	0.9 (0.1–2.8)	14	We	5.1	2.2 (1.8–2.5)	2.8	2
						Be	0.9	1.0	0.9	1

**Table 3 microorganisms-09-00317-t003:** Pearson correlation analyses for each variable between the SML and the SSW for Arctic and Antarctic seawaters. Values in bold are significant. Acronyms of variables are shown in Table 1 and Table 2. -: no data.

Variables	*r*	Arctic (*p*)	*n*	*r*	Antarctic (*p*)	*n*
VA	0.607	**<0.006**	19	0.689	**<0.001**	19
PA	0.782	**<0.001**	19	0.823	**<0.0001**	19
VPR	0.423	**<0.05**	19	0.586	**<0.008**	19
PHP	0.805	**<0.0001**	19	0.625	**<0.0042**	18
VPL	0.486	>0.05	8	0.621	>0.05	6
RLC	0.221	>0.05	8	−0.011	>0.05	6
%VMM	−0.139	>0.05	8	−0.795	**<0.05**	6
HNF	0.812	**<0.0001**	15	0.406	>0.05	5
PNF	0.804	**<0.0001**	14	0.018	>0.05	5
T	0.929	**<0.0001**	19	-	-	-
Sal	0.994	**<0.0001**	19	-	-	-
DOC	0.582	**<0.01**	18	-	-	-
PO_4_	0.718	**<0.001**	17	0.771	**<0.001**	13
SiO_4_	0.702	**<0.001**	18	0.892	**<0.0001**	13
NO_3_+NO_2_	0.949	**<0.0001**	18	0.605	**<0.03**	12
NH_4_	0.629	**<0.001**	19	0.921	**<0.01**	5

**Table 4 microorganisms-09-00317-t004:** ANOVA analysis for variables measured in the SML and SSW, as well as in the air between Arctic and Antarctic systems. df: degree of freedom; *F*: Fisher coefficient; *p*: significance level values in bold. * Significant higher values in Arctic than in Antarctic waters; * significant higher values in Antarctic than in Arctic waters. ns: no significant values. For variable acronyms and units, see Table 1 and Table 2.

Variable	Water/Air	df	*F*	*p*	Value Comparison
Temperature	SSW	36	**3.72**	**0.05**	*
Salinity	SSW	36	**21.94**	**<0.0001**	*
DOC	SSW	34	**13.2**	**<0.0009**	*
	EF	28	0.780	0.345	ns
PO_4_	SML	29	**91.11**	**<0.00001**	*
	SSW	34	**121.36**	**<0.00001**	*
	EF	29	0.66	0.42	ns
SiO_4_	SML	29	**195.75**	**<0.00001**	*
	SSW	34	**777.68**	**<0.00001**	*
	EF	28	0.15	0.70	ns
NO_3_+NO_2_	SML	28	**112.60**	**<0.00001**	*
	SSW	33	**114.91**	**<0.00001**	*
	EF	24	1.69	0.206	ns
NH_4_	SML	24	1.84	0.190	ns
	SSW	28	**70.58**	**0.003**	*
Wind	air	36	0.70	0.391	ns
UVB	air	21	**12.51**	**0.0021**	*
	EF	37	0.16	0.693	ns
VA	SML	37	1.81	0.186	ns
	SSW	37	2.05	0.161	ns
	EF	37	0.67	0.418	ns
VPR	SML	37	**10.46**	**0.003**	*
	SSW	37	**14.52**	**0.0005**	*
	EF	37	0.06	0.805	ns
PA	SML	37	**5.49**	**0.025**	*
	SSW	37	**18.21**	**0.0001**	*
	EF	37	1.09	0.301	ns
PHP	SML	37	0.97	0.331	ns
	SSW	37	**5.80**	**0.021**	*
	EF	13	**5.98**	**0.031**	*
VPL	SML	13	**95.44**	**<0.00001**	*
	SSW	13	**41.47**	**<0.0001**	*
	EF	13	1.46	0.251	ns
RLC	SML	13	1.69	0.219	ns
	SSW	13	**11.08**	**0.006**	*
	EF	13	0.83	0.380	ns
%VMM	SML	13	**14.49**	**0.003**	*
	SSW	13	**21.47**	**0.0006**	*
	EF	19	0.37	0.552	ns
HNF	SML	19	0.27	0.609	ns
	SSW	19	0.003	0.955	ns
	EF	18	**5.22**	**0.036**	*
PNF	SML	19	0.23	0.636	ns
	SSW	19	**20.27**	**0.0003**	*

**Table 5 microorganisms-09-00317-t005:** Enrichment factor (EF) of abundance of viruses (VA), prokaryotes (PA), rate of lysed cells (RLC), and concentrations of dissolved organic carbon (DOC) and phosphate (PO_4_) under different wind speeds for both polar systems. Significantly different values (Student’s *t*-test, *p* < 0.05) between low (≤5 m s^−1^) and high (>6 m s^−1^) wind speeds are in bold; -: no data.

Site	Wind Speed	*n*	EF–VA	*n*	EF–PA	*n*	EF–RLC	*n*	EF–DOC	*n*	EF–PO_4_
**AR**	Range: 1.2–7.9										
	≤5 m s^−1^	9	**3.0 ± 1.5**	9	1.4 ± 0.4	4	**47.9 ± 21.2**	9	**1.6 ± 0.5**	8	1.4 ± 0.2
	5–6 m s^−1^	4	2.4 ± 0.4	4	1.2 ± 0.1	1	3.6 ± 0.0	3	1.3 ± 0.3	4	1.3 ± 0.1
	>6 m s^−1^	6	0.9 ± 0.1	6	0.9 ± 0.1	3	7.5 ± 3.4	6	0.9 ± 0.1	5	0.9 ± 0.3
**AN**	Range: 2.6–9.9										
	≤5 m s^−1^	7	2.6 ± 1.4	7	**1.6 ± 0.2**	1	6.2 ± 0.00		-	7	1.3 ± 0.2
	5–6 m s^−1^	5	3.7 ± 1.3	5	1.1 ± 0.04	3	5.7 ± 0.6		-	1	1.0 ± 0.0
	>6 m s^−1^	7	1.9 ± 0.3	9	1.1 ± 0.1	2	5.2 ± 0.2		-	6	1.0 ± 0.03

## Data Availability

All data is reported in the present article.

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
