# Peer review of "Enhanced Viral Activity in the Surface Microlayer of the Arctic and Antarctic Oceans"

_microorganisms, 2021, doi:10.3390/microorganisms9020317_

Round 1
Reviewer 1 Report
Review for manuscript microorganisms-1083791
The manuscript „Enhanced viral activity in the surface microlayer of the Arctic and Antarctic Oceans” by Vaqué et al describes various parameters (e.g. environmental, viral, prokaryote and flagellate abundances, virus-induced mortality etc.) for the sea-surface microlayer and SSW for different Arctic and Antarctic stations. The manuscript is well-written, and the dataset is valuable, because knowledge on viruses in the SML is scarce, which is especially true for polar regions. I have some suggestions for improvements as follow:
Abstract: Would be good to have a final statement about the meaning of the results for the ‘bigger picture’
L73: “no single study reported on Antarctic SML”. This is not true. Please consider adding the following refs or re-phrase that you refer to viruses only in your statement.
doi
10.3389/fmicb.2020.571983
10.1099/ijsem.0.004240
10.1099/ijsem.0.001202
Figure 1: there is quite some unused space in this figure, the maps could be bigger.
L.125/126: Please clarify what is collected in these SML blanks: the ultrapure water used for cleaning, or 500 ml SML collected with the plate thereafter?
L.132: As far as I understand, temperature in SML and SSW has been measured after sample collection and not in situ in the different layers? Please clarify in the text.
L.146: If wind speed has been measured using a weather station on the R/V, it does not correspond to the height at sea surface level and should be converted accordingly.
L.151: I would separate microbes from viruses, as viruses are biological entities but not living microorganisms. Please check throughout the manuscript.
L.152: Were biological replicates measured for viruses and prokaryote abundance in flow cytometry?
L.239: virus-induced mortality? Or mortality in general?
L.245f. very important to add if this is really in situ temp. or temp. measured in collected SML and SSW. If in collected SML, how much time was between sampling and measurement? Also, it is interesting that SML temp was higher in SML compared to SSW, because this is not always the case, e.g. in the tropics, please see doi: 10.1038/s41598-018-29869-7
Where SML temp was slightly cooler than SSW. Perhaps this is worth a discussion, as temperature could certainly affect viral activity.
L.270: The concept of enrichment factor (EF) needs to be explained in the methods.
Table 1: if the EF is bold, i.e. significant, what does this exactly mean? What is compared here? Please add an explanation.
Figure 2: What does the * mean?
L.309/310: Isn’t it more common to use the term “virus-like particle”?
L.316/317: The correlations mentioned here: could they exist because your reference depth at 0.1 m was chosen somewhat shallow compared to other studies? Please discuss.
Table 2 &3: significance is for comparison SML/SSW or Arctic/Antarctic? I assume SML and SSW, but it could be stated more clearly.
L.329/330: compared to Arctic? Please check throughout the manuscript that comparisons are clearly explained.
L.336-345: Were there any correlations between HNF or PNF and viruses? Because in principle the virus counts could also refer to eukaryotic viruses, not only bacteriophages.
L.404: is not a full sentence. Please write complete words for “nd”,
L.430: “Prokaryotes were the main host…”. How do you know? You did not check for eukaryotic viruses.
L.431: please use SML consistently throughout the manuscript.
L.435: is the depletion of PNF in SML a sign for photoinhibition? (also relevant for line 462)
L.441: also consider viral deposition from the atmosphere and maybe add in that context doi: 10.1038/s41396-017-0042-4
L.442: is this EF range for the prokaryotes or the viruses? Would be good to see both ranges in comparison.
L.504: microbial loop.
I miss the term viral shunt when it comes to carbon release due to viral lysis and some recent literature, e.g. doi: 10.1038/s41579-019-0270-x
Reviewer 2 Report
This manuscript provides an in-depth analysis of the surface microlayer and the subsurface waters of different sampling locations in the Arctic and Antarctic. Overall, the paper provides new, interesting and well-described data. It is in general suitable for publication, however, the following aspects should be addressed beforehand:
- The manuscript should be proofread by a native speaker, as there are numerous grammatical and spelling mistakes and odd sounding sentences.
- Line 25: Explain the abbreviation “DOC”
- Table 1: I suggest splitting table 1 into two separate tables, one for the physicochemical parameters and one for the microbial parameters.
- Figure 2: Define in the legend what is represented by the box-whisker plots (mean or median / percentiles?). It would be helpful to also define “enrichment factor” in the legend.
- Figure 3: The legend needs to state what the bars and whiskers represent. It says measurements in duplicates (which I assume are, for example, dark red and light red bars) but each bar also has a whisker. What is the white box on top of some bars in E and F?
- Figure 4: Describe in the legend how the linear regression curves and p values were calculated (it is stated in the Methods but should also be stated in the legend).
